# Rethinking the Value of Labels for Improving Class-Imbalanced Learning

**Yuzhe Yang**
EECS
Massachusetts Institute of Technology
`yuzhe@mit.edu`

**Zhi Xu**
EECS
Massachusetts Institute of Technology
`zhixu@mit.edu`

## Abstract

Real-world data often exhibits long-tailed distributions with heavy class imbalance, posing great challenges for deep recognition models. We identify a persisting dilemma on the *value of labels* in the context of imbalanced learning: on the one hand, supervision from labels typically leads to better results than its unsupervised counterparts; on the other hand, heavily imbalanced data naturally incurs "label bias" in the classifier, where the decision boundary can be drastically altered by the majority classes. In this work, we systematically investigate these two facets of labels. We demonstrate, theoretically and empirically, that class-imbalanced learning can significantly benefit in both semi-supervised and self-supervised manners. Specifically, we confirm that (1) positively, imbalanced labels are valuable: given more unlabeled data, the original labels can be leveraged with the extra data to reduce label bias in a semi-supervised manner, which greatly improves the final classifier; (2) negatively however, we argue that imbalanced labels are not useful always: classifiers that are first pre-trained in a self-supervised manner consistently outperform their corresponding baselines. Extensive experiments on large-scale imbalanced datasets verify our theoretically grounded strategies, showing superior performance over previous state-of-the-arts. Our intriguing findings highlight the need to rethink the usage of imbalanced labels in realistic long-tailed tasks. Code is available at `https://github.com/YyzHarry/imbalanced-semi-self`.

## 1 Introduction

Imbalanced data is ubiquitous in the real world, where large-scale datasets often exhibit long-tailed label distributions [1,5,24,33]. In particular, for critical applications related to safety or health, such as autonomous driving and medical diagnosis, the data are by their nature heavily imbalanced. This posts a major challenge for modern deep learning frameworks [2,5,10,20,53], where even with specialized techniques such as data re-sampling approaches [2,5,41] or class-balanced losses [7,13,26], significant performance drops still remain under extreme class imbalance. In order to further tackle the challenge, it is hence vital to understand the different characteristics incurred by class-imbalanced learning.

Yet, distinct from balanced data, the labels in the context of imbalanced learning play a surprisingly controversial role, which leads to a persisting dilemma on the *value of labels*: (1) On the one hand, learning algorithms with supervision from labels typically result in more accurate classifiers than their unsupervised counterparts, demonstrating the *positive* value of labels; (2) On the other hand, however, imbalanced labels naturally impose "label bias" during learning, where the decision boundary can be significantly driven by the majority classes, demonstrating the *negative* impact of labels. Hence, the imbalanced label is seemingly a double-edged sword. Naturally, a fundamental question arises:

> *How to maximally exploit the value of labels to improve class-imbalanced learning?*

In this work, we take initiatives to *systematically* decompose and analyze the two facets of imbalanced labels. As our key contributions, we demonstrate that the positive *and* the negative viewpoint of the

dilemma are indeed both enlightening: they can be effectively exploited, in semi-supervised and self-supervised manners respectively, to significantly improve the state-of-the-art.

On the positive viewpoint, we argue that imbalanced labels are indeed valuable. Theoretically, via a simple Gaussian model, we show that extra unlabeled data benefits imbalanced learning: we obtain a close estimate with high probability that increases exponentially in the amount of unlabeled data, even when unlabeled data are also (highly) imbalanced. Inspired by this, we confirm empirically that by leveraging the label information, class-imbalanced learning can be substantially improved by employing a simple pseudo-labeling strategy, which alleviates the label bias with extra data in a *semi-supervised* manner. Regardless of the imbalanceness on both the labeled and unlabeled data, superior performance is established consistently across various benchmarks, signifying the valuable supervision from the imbalanced labels that leads to substantially better classifiers.

On the negative viewpoint, we demonstrate that imbalanced labels are not advantageous all the time. Theoretically, via a high-dimensional Gaussian model, we show that if given informative representations learned without using labels, with high probability depending on the imbalanceness, we obtain classifier with exponentially small error probability, while the raw classifier always has constant error. Motivated by this, we verify empirically that by abandoning label at the beginning, classifiers that are first pre-trained in a *self-supervised* manner consistently outperform their corresponding baselines, regardless of settings and base training techniques. Significant improvements on large-scale datasets reveal that the biased label information can be greatly compensated through natural self-supervision.

Overall, our intriguing findings highlight the need to rethink the usage of imbalanced labels in realistic imbalanced tasks. With strong performance gains, we establish that not only the positive viewpoint but also the negative one are both promising directions for improving class-imbalanced learning.

**Contributions. (i)** We are the first to systematically analyze imbalanced learning through two facets of imbalanced label, validating and exploiting its value in novel semi- and self-supervised manners. **(ii)** We demonstrate, theoretically and empirically, that using unlabeled data can substantially boost imbalanced learning through semi-supervised strategies. **(iii)** Further, we introduce self-supervised pre-training for class-imbalanced learning without using any extra data, exhibiting both appealing theoretical interpretations and new state-of-the-art on large-scale imbalanced benchmarks.

## 2 Imbalanced Learning with Unlabeled Data

As motivated, we explore the positive value of labels. Naturally, we consider scenarios where extra unlabeled data is available and hence, the limited labeling information is critical. Through a simple theoretical model, we first build intuitions on how different ingredients of the originally imbalanced data and the extra data affect the overall learning process. With a clearer picture in mind, we design comprehensive experiments to confirm the efficacy of this direction on boosting imbalanced learning.

### 2.1 Theoretical Motivation

Consider a binary classification problem with the data generating distribution $P_{XY}$ being a mixture of two Gaussians. In particular, the label $Y$ is either positive (+1) or negative (-1) with equal probability (i.e., 0.5). Condition on $Y = +1$, $X|Y = +1 \sim \mathcal{N}(\mu_1, \sigma^2)$ and similarly, $X|Y = -1 \sim \mathcal{N}(\mu_2, \sigma^2)$. Without loss of generality, let $\mu_1 > \mu_2$. It is straightforward to verify that the optimal Bayes's classifier is $f(x) = \text{sign}(x - \frac{\mu_1+\mu_2}{2})$, i.e., classify $x$ as +1 if $x > (\mu_1 + \mu_2)/2$. Therefore, in the following, we measure our ability to learn $(\mu_1 + \mu_2)/2$ as a proxy for performance.

Suppose that a base classifier $f_B$, trained on imbalanced training data, is given. We consider the case where extra unlabeled data $\{\tilde{X}_i\}_i^{\tilde{n}}$ (potentially also imbalanced) from $P_{XY}$ are available, and study how this affects our performance with the label information from $f_B$. Precisely, we create pseudo-label for $\{\tilde{X}_i\}_i^{\tilde{n}}$ using $f_B$. Let $\{\tilde{X}_i^+\}_{i=1}^{\tilde{n}_+}$ be the set of unlabeled data whose pseudo-label is +1; similarly let $\{\tilde{X}_i^-\}_{i=1}^{\tilde{n}_-}$ be the negative set. Naturally, when the training data is imbalanced, $f_B$ is likely to exhibit different accuracy for different class. We model this as follows. Consider the case where pseudo-label is +1 and let $\{I_i^+\}_{i=1}^{\tilde{n}_+}$ be the indicator that the $i$-th pseudo-label is correct, i.e., if $I_i^+ = 1$, then $\tilde{X}_i^+ \sim \mathcal{N}(\mu_1, \sigma^2)$ and otherwise $\tilde{X}_i^+ \sim \mathcal{N}(\mu_2, \sigma^2)$. We assume $I_i^+ \sim \text{Bernoulli}(p)$, which means $f_B$ has an accuracy of $p$ for the positive class. Analogously, we define $\{I_i^-\}_{i=1}^{\tilde{n}_-}$, i.e., if $I_i^- = 1$, then $\tilde{X}_i^- \sim \mathcal{N}(\mu_2, \sigma^2)$ and otherwise $\tilde{X}_i^- \sim \mathcal{N}(\mu_1, \sigma^2)$. Let $I_i^- \sim \text{Bernoulli}(q)$, which

means $f_B$ has an accuracy of $q$ for the negative class. Denote by $\Delta \triangleq p - q$ the imbalance in accuracy. As mentioned, we aim to learn $(\mu_1 + \mu_2)/2$ with the above setup, via the extra unlabeled data. It is natural to construct our estimate as $\hat{\theta} = \frac{1}{2}\big(\sum_{i=1}^{\tilde{n}_+} \tilde{X}_i^+/\tilde{n}_+ + \sum_{i=1}^{\tilde{n}_-} \tilde{X}_i^-/\tilde{n}_-\big)$. Then, we have:

**Theorem 1.** *Consider the above setup. For any $\delta > 0$, with probability at least $1 - 2e^{-\frac{2\delta^2}{9\sigma^2} \cdot \frac{\tilde{n}_+ + \tilde{n}_-}{\tilde{n}_- + \tilde{n}_+}} -$ $2e^{-\frac{8\tilde{n}_+ \delta^2}{9(\mu_1 - \mu_2)^2}} - 2e^{-\frac{8\tilde{n}_- \delta^2}{9(\mu_1 - \mu_2)^2}}$ our estimates $\hat{\theta}$ satisfies*

$$\left| \hat{\theta} - (\mu_1 + \mu_2)/2 - \Delta(\mu_1 - \mu_2)/2 \right| \leq \delta.$$

**Interpretation.** The result illustrates several interesting aspects. (1) *Training data imbalance affects the accuracy of our estimation.* For heavily imbalanced training data, we expect the base classifier to have a large difference in accuracy between major and minor classes. That is, the more imbalanced the data is, the larger the gap $\Delta$ would be, which influences the closeness between our estimate and desired value $(\mu_1 + \mu_2)/2$. (2) *Unlabeled data imbalance affects the probability of obtaining such a good estimation.* For a reasonably good base classifier, we can roughly view $\tilde{n}_+$ and $\tilde{n}_-$ as approximations for the number of actually positive and negative data in unlabeled set. For term $2\exp(-\frac{2\delta^2}{9\sigma^2} \cdot \frac{\tilde{n}_+ + \tilde{n}_-}{\tilde{n}_- + \tilde{n}_+})$, note that $\frac{\tilde{n}_+ + \tilde{n}_-}{\tilde{n}_- + \tilde{n}_+}$ is maximized when $\tilde{n}_+ = \tilde{n}_-$, i.e., balanced unlabeled data. For terms $2\exp(-\frac{8\tilde{n}_+ \delta^2}{9(\mu_1 - \mu_2)^2})$ and $2\exp(-\frac{8\tilde{n}_- \delta^2}{9(\mu_1 - \mu_2)^2})$, if the unlabeled data is heavily imbalanced, then the term corresponding to the minor class dominates and can be moderately large. Our probability of success would be higher with balanced data, but in any case, more unlabeled data is *always* helpful.

## 2.2 Semi-Supervised Imbalanced Learning Framework

Our theoretical findings show that pseudo-label (and hence the label information in training data) can be helpful in imbalanced learning. The degree to which this is useful is affected by the imbalanceness of the data. Inspired by these, we systematically probe the effectiveness of unlabeled data and study how it can improve realistic imbalanced task, especially with varying degree of imbalanceness.

**Semi-Supervised Imbalanced Learning.** To harness the unlabeled data for alleviating the inherent imbalance, we propose to adopt the classic *self-training* framework, which performs semi-supervised learning (SSL) by generating *pseudo-labels* for unlabeled data. Precisely, we obtain an intermediate classifier $f_{\hat{\theta}}$ using the original imbalanced dataset $\mathcal{D}_L$, and apply it to generate pseudo-labels $\hat{y}$ for unlabeled data $\mathcal{D}_U$. The data and pseudo-labels are combined to learn a final model $f_{\hat{\theta}_f}$ by minimizing a loss function as $\mathcal{L}(\mathcal{D}_L, \theta) + \omega \mathcal{L}(\mathcal{D}_U, \theta)$, where $\omega$ is the unlabeled weight. This procedure seeks to remodel the class distribution with $\mathcal{D}_U$, obtaining better class boundaries especially for tail classes.

We remark that besides self-training, more advanced SSL techniques can be easily incorporated into our framework by modifying only the loss function, which we will study later. As we do not specify the learning strategy of $f_{\hat{\theta}}$ and $f_{\hat{\theta}_f}$, the semi-supervised framework is also compatible with existing class-imbalanced learning methods. Accordingly, we demonstrate the value of unlabeled data — a simple self-training procedure can lead to substantially better performance for imbalanced learning.

**Experimental Setup.** We conduct thorough experiments on artificially created long-tailed versions of CIFAR-10 [7] and SVHN [36], which naturally have their unlabeled part with similar distributions: 80 Million Tiny Images [48] for CIFAR-10, and SVHN's own extra set [36] with labels removed for SVHN. Following [7, 11], the class *imbalance ratio* $\rho$ is defined as the sample size of the most frequent (head) class divided by that of the least frequent (tail) class. Similarly for $\mathcal{D}_U$, we define the *unlabeled imbalance ratio* $\rho_U$ in the same way. More details of datasets are reported in Appendix D.

For long-tailed dataset with a fixed $\rho$, we augment it with 5 times more unlabeled data, denoted as $\mathcal{D}_U@5\mathrm{x}$. As we seek to study the effect of unlabeled imbalance ratio, the total size of $\mathcal{D}_U@5\mathrm{x}$ is fixed, where we vary $\rho_U$ to obtain corresponding imbalanced $\mathcal{D}_U$. We select standard cross-entropy (CE) training, and a recently proposed state-of-the-art imbalanced learning method LDAM-DRW [7] as baseline methods. We follow [7, 25, 33] to evaluate models on corresponding balanced test datasets.

### 2.2.1 Main Results

**CIFAR-10-LT & SVHN-LT.** Table 1 summarizes the results on two long-tailed datasets. For each $\rho$, we vary the type of class imbalance in $\mathcal{D}_U$ to be uniform ($\rho_U = 1$), half as labeled ($\rho_U = \rho/2$), same ($\rho_U = \rho$), and doubled ($\rho_U = 2\rho$). As shown in the table, the SSL scheme can consistently and

Table 1: Top-1 test errors (%) of ResNet-32 on long-tailed CIFAR-10 and SVHN. We compare SSL using 5x unlabeled data ($\mathcal{D}_U$@5x) with corresponding supervised baselines. Imbalanced learning can be drastically improved with unlabeled data, which is consistent across different $\rho_U$ and learning strategies.

(a) CIFAR-10-LT

| Imbalance Ratio ($\rho$) | 100 | | | | 50 | | | | 10 | | | |
|---|---|---|---|---|---|---|---|---|---|---|---|---|
| $\mathcal{D}_U$ Imbalance Ratio ($\rho_U$) | 1 | $\rho/2$ | $\rho$ | $2\rho$ | 1 | $\rho/2$ | $\rho$ | $2\rho$ | 1 | $\rho/2$ | $\rho$ | $2\rho$ |
| CE | 29.64 | | | | 25.19 | | | | 13.61 | | | |
| CE + $\mathcal{D}_U$@5x | **17.48** | 18.42 | 18.74 | 20.06 | **16.79** | 16.88 | 18.36 | 19.94 | **10.22** | 10.48 | 10.86 | 11.04 |
| LDAM-DRW [7] | 22.97 | | | | 19.06 | | | | 11.84 | | | |
| LDAM-DRW + $\mathcal{D}_U$@5x | **14.96** | 15.18 | 15.33 | 15.55 | **14.33** | 14.70 | 14.93 | 15.24 | 8.72 | **8.24** | 8.68 | 8.97 |

(b) SVHN-LT

| Imbalance Ratio ($\rho$) | 100 | | | | 50 | | | | 10 | | | |
|---|---|---|---|---|---|---|---|---|---|---|---|---|
| $\mathcal{D}_U$ Imbalance Ratio ($\rho_U$) | 1 | $\rho/2$ | $\rho$ | $2\rho$ | 1 | $\rho/2$ | $\rho$ | $2\rho$ | 1 | $\rho/2$ | $\rho$ | $2\rho$ |
| CE | 19.98 | | | | 17.50 | | | | 11.46 | | | |
| CE + $\mathcal{D}_U$@5x | **13.02** | 13.73 | 14.65 | 15.04 | **13.07** | 13.36 | 13.16 | 14.54 | **10.01** | 10.20 | 10.06 | 10.71 |
| LDAM-DRW [7] | 16.66 | | | | 14.59 | | | | 10.27 | | | |
| LDAM-DRW + $\mathcal{D}_U$@5x | **11.32** | 11.70 | 11.92 | 12.78 | **10.98** | 11.14 | 11.26 | 11.51 | 8.94 | 9.08 | **8.70** | 9.35 |

substantially improve existing techniques across different $\rho$. Notably, under extreme class imbalance ($\rho = 100$), using unlabeled data can lead to +10% on CIFAR-10-LT, and +6% on SVHN-LT.

**Imbalanced Distribution in Unlabeled Data.** As indicated by Theorem 1, unlabeled data imbalance affects the learning of final classifier. We observe in Table 1 that gains indeed vary under different $\rho_U$, with smaller $\rho_U$ (i.e., more balanced $\mathcal{D}_U$) leading to larger gains. Interestingly however, as the original dataset becomes more balanced, the benefits from $\mathcal{D}_U$ tend to be similar across different $\rho_U$.

**Qualitative Results.** To further understand the effect of unlabeled data, we visualize representations learned with vanilla CE (Fig. 1a) and with SSL (Fig. 1b) using t-SNE [34]. The figures show that imbalanced training set can lead to poor class separation, particularly for tail classes, which results in mixed class boundary during class-balanced inference. In contrast, by leveraging unlabeled data, the boundary of tail classes can be better shaped, leading to clearer separation and better performance.

**Summary.** Consistently across various settings, class-imbalanced learning tasks benefit *greatly* from additional unlabeled data. The superior performance obtained demonstrates the positive value of imbalanced labels as being capable of exploiting the unlabeled data for extra benefits.

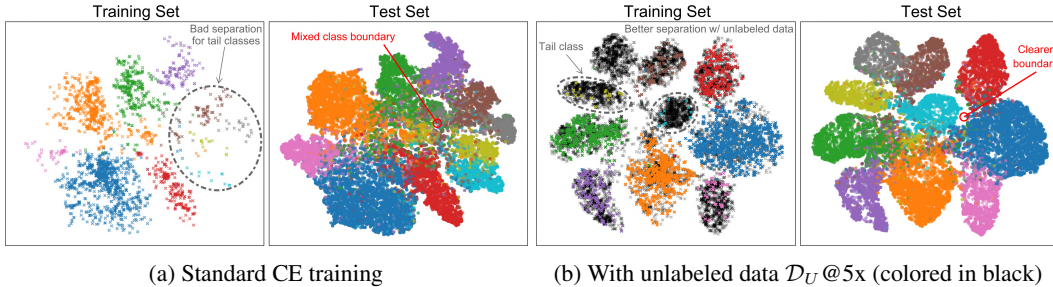

(a) Standard CE training        (b) With unlabeled data $\mathcal{D}_U$@5x (colored in black)

Figure 1: t-SNE visualization of training & test set on SVHN-LT. Using unlabeled data helps to shape clearer class boundaries and results in better class separation, especially for the tail classes.

### 2.2.2 Further Analysis and Ablation Studies

**Different Semi-Supervised Methods (Appendix E.1).** In addition to the simple pseudo-label, we select more advanced SSL techniques [35, 46] and explore the effect of different methods under the

imbalanced settings. In short, all SSL methods can achieve remarkable performance gains over the supervised baselines, with more advanced SSL method enjoying larger improvements in general.

**Generalization on Minority Classes (Appendix E.2).** Besides the reported top-1 accuracy, we further study generalization on each class with and without unlabeled data. We show that while all classes can obtain certain improvements, the minority classes tend to exhibit larger gains.

**Unlabeled & Labeled Data Amount (Appendix E.3 & E.4).** Following [39], we investigate how different amounts of $\mathcal{D}_U$ as well as $\mathcal{D}_L$ can affect our SSL approach in imbalanced learning. We find that larger $\mathcal{D}_U$ or $\mathcal{D}_L$ often brings higher gains, with gains gradually diminish as data amount grows.

# 3   A Closer Look at Unlabeled Data under Class Imbalance

With significant boost in performance, we confirm the value of imbalanced labels with extra unlabeled data. Such success naturally motivates us to dig deeper into the techniques and investigate whether SSL is the solution to practical imbalanced data. Indeed, in the balanced case, SSL is known to have issues in certain scenarios when the unlabeled data is not ideally constructed. Techniques are often sensitive to the relevance of unlabeled data, and performance can even degrade if the unlabeled data is largely mismatched [39]. The situation becomes even more complicated when imbalance comes into the picture. The relevant unlabeled data could also exhibit long-tailed distributions. With this, we aim to further provide an informed message on the utility of semi-supervised techniques.

**Data Relevance under Imbalance.** We construct sets of unlabeled data with the same imbalance ratio as the training data but varying relevance. Specifically, we mix the original unlabeled dataset with irrelevant data, and create unlabeled datasets with varying data relevance ratios (detailed setup can be found in Appendix D.2). Fig. 2 shows that in imbalanced learning, adding unlabeled data from mismatched classes can actually *hurt* performance. The relevance has to be as high as $60\%$ to be effective, while better results could be obtained without unlabeled data at all when it is only moderately relevant. The observations are consistent with the balanced cases [39].

**Varying $\rho_U$ under Sufficient Data Relevance.** Furthermore, even with enough relevance, what will happen if the relevant unlabeled data is (heavily) long-tailed? As presented in Fig. 3, for a fixed relevance, the higher $\rho_U$ of the relevant data is, the higher the test error. In this case, to be helpful, $\rho_U$ cannot be larger than 50 (which is the imbalance ratio of the training data). This highlights that unlike traditional setting, the imbalance of the unlabeled data imposes an additional challenge.

**Why Do These Matter.** The observations signify that semi-supervised techniques should be applied with care in practice for imbalanced learning. When it is readily to obtain relevant unlabeled data of each class, they are particularly powerful as we demonstrate. However, certain practical applications, especially those extremely imbalanced, are situated at the worst end of the spectrum. For example, in medical diagnosis, positive samples are always scarce; even with access to more "unlabeled" medical records, positive samples remain sparse, and the confounding issues (e.g., other disease or symptoms) undoubtedly hurt relevance. Therefore, one would expect the imbalance ratio of the unlabeled data to be higher, if not lower, than the training data in these applications.

To summarize, unlabeled data are useful. However, semi-supervised learning alone is *not* sufficient to solve the imbalanced problem. Other techniques are needed in case the application does not allow constructing meaningful unlabeled data, and this, exactly motivates our subsequent studies.

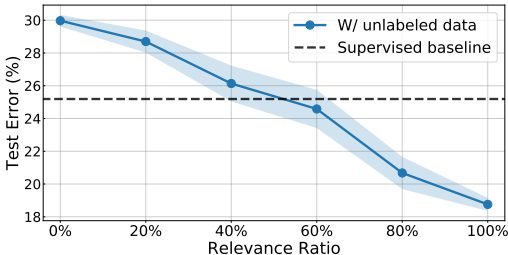

Figure 2: Test errors of different unlabeled data relevance ratios on CIFAR-10-LT with $\rho = 50$. We fix $\rho_U = 50$ for the relevant unlabeled data.

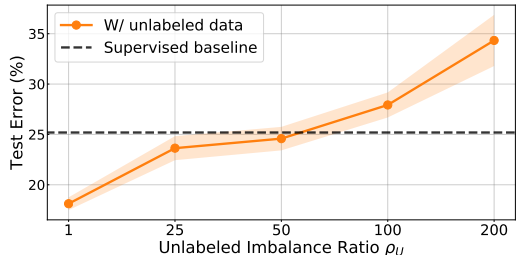

Figure 3: Test errors of different $\rho_U$ of relevant unlabeled data on CIFAR-10-LT with $\rho = 50$. We fix the unlabeled data relevance ratio as $60\%$.

# 4 Imbalanced Learning from Self-Supervision

The previous studies naturally motivate our next quest: can the negative viewpoint of the dilemma, i.e., the imbalanced labels introduce bias and hence are "unnecessary", be successfully exploited as well to advance imbalanced learning? In answering this, our goal is to seek techniques that can be broadly applied without extra data. Through a theoretical model, we first justify the usage of self-supervision in the context of imbalanced learning. Extensive experiments are then designed to verify its effectiveness, proving that thinking through the negative viewpoint is indeed promising as well.

## 4.1 Theoretical Motivation

We start with another inspiring model to study how imbalanced learning benefits from self-supervision. Consider $d$-dimensional binary classification with data generating distribution $P_{XY}$ being a mixture of Gaussians. In particular, the label $Y = +1$ with probability $p_+$ while $Y = -1$ with probability $p_- = 1 - p_+$. Let $p_- \geq 0.5$, i.e., major class is negative. Condition on $Y = +1$, $X$ is a $d$-dimensional isotropic Gaussian, i.e., $X|Y = +1 \sim \mathcal{N}(0, \sigma_1^2 \mathbf{I}_d)$. Similarly, $X|Y = -1 \sim \mathcal{N}(0, \beta \sigma_1^2 \mathbf{I}_d)$ for some constant $\beta > 3$, i.e., the negative samples have larger variance. The training data, $\{(X_i, Y_i)\}_{i=1}^N$, could be highly imbalanced, and we denote by $N_+$ & $N_-$ as number of positive & negative samples.

To develop our intuition, we consider learning a linear classifier with and without self-supervision. In particular, consider the class of linear classifiers $f(x) = \text{sign}(\langle \theta, \text{feature} \rangle + b)$, where feature would be the raw input $X$ in standard training, and for the self-supervised learning, feature would be $Z = \psi(X)$ for some representation $\psi$ learned through a self-supervised task. For convenience, we consider the case where the intercept $b \geq 0$. We assume a properly designed black-box self-supervised task so that the learned representation is $Z = k_1 ||X||_2^2 + k_2$, where $k_1, k_2 > 0$. Precisely, this means that we have access to the new features $Z_i$ for the $i$-th data after the black-box self-supervised step, without knowing explicitly what the transformation $\psi$ is. Finally, we measure the performance of a classifier $f$ using the standard error probability: $\text{err}_f = \mathbb{P}_{(X,Y)\sim P_{XY}}(f(X) \neq Y)$.

**Theorem 2.** *Let $\Phi$ be the CDF of $\mathcal{N}(0,1)$. For any linear classifier of the form $f(X) = \text{sign}(\langle \theta, X \rangle + b)$ where $b > 0$, the error probability satisfies:* $\text{err}_f = p_+ \Phi\left(-\frac{b}{||\theta||_2 \sigma_1}\right) + p_- \Phi\left(\frac{b}{||\theta||_2 \sqrt{\beta} \sigma_1}\right) \geq \frac{1}{4}$.

Theorem 2 states that for standard training, regardless of whether the training data is imbalanced or not, the linear classifier cannot have an accuracy $\geq 3/4$. This is rather discouraging for such a simple case. However, we show that self-supervision and training on the resulting $Z$ provides a better classifier. Consider the same linear class $f(x) = \text{sign}(\langle \theta, \text{feature} \rangle + b)$, $b > 0$ and following explicit classifier with feature $Z = \psi(X)$: $f_{ss}(X) = \text{sign}(-Z + b)$, $b = \frac{1}{2}\left(\frac{\sum_{i=1}^N \mathbf{1}_{\{Y_i=+1\}} Z_i}{N_+} + \frac{\sum_{i=1}^N \mathbf{1}_{\{Y_i=-1\}} Z_i}{N_-}\right)$. The next theorem shows a high probability error bound for the performance of this linear classifier.

**Theorem 3.** *Consider the linear classifier with self-supervised learning, $f_{ss}$. For any $\delta \in \left(0, \frac{\beta-1}{\beta+1}\right)$, we have that with probability at least $1 - 2e^{-N_- d\delta^2/8} - 2e^{-N_+ d\delta^2/8}$, the classifier satisfies*

$$
\text{err}_{f_{ss}} \leq \begin{cases} p_+ e^{-d \cdot \frac{(\beta-1-(1+\beta)\delta)^2}{32}} + p_- e^{-d \cdot \frac{(\beta-1-(1+\beta)\delta)^2}{32\beta^2}}, & \text{if } \delta \in \left[\frac{\beta-3}{\beta+1}, \frac{\beta-1}{\beta+1}\right); \\ p_+ e^{-d \cdot \frac{(\beta-1-(1+\beta)\delta)}{16}} + p_- e^{-d \cdot \frac{(\beta-1-(1+\beta)\delta)^2}{32\beta^2}}, & \text{if } \delta \in \left(0, \frac{\beta-3}{\beta+1}\right). \end{cases}
$$

**Interpretation.** Theorem 3 implies the following interesting observations. By first abandoning imbalanced labels and learning an informative representation via self-supervision, (1) *With high probability, we obtain a satisfying classifier $f_{ss}$, whose error probability decays exponentially on the dimension $d$.* The probability of obtaining such a classifier also depends exponentially on $d$ and the number of data. These are rather appealing as modern data is of extremely high dimension. That is, even for imbalanced data, one could obtain a good classifier with proper self-supervised training; (2) *Training data imbalance affects our probability of obtaining such a satisfying classifier.* Precisely, given $N$ data, if it is highly imbalanced with an extremely small $N_+$, then the term $2\exp(-N_+ d\delta^2/8)$ could be moderate and dominate $2\exp(-N_- d\delta^2/8)$. With more or less balanced data (or just more data), our probability of success increases. Nevertheless, as the dependence is exponential, even for imbalanced training data, self-supervised learning can still help to obtain a satisfying classifier.

Table 2: Top-1 test error rates (%) of ResNet-32 on long-tailed CIFAR-10 and CIFAR-100. Using SSP, we consistently improve different imbalanced learning techniques, and achieve the best performance.

| Dataset | CIFAR-10-LT | | | CIFAR-100-LT | | |
|---|---|---|---|---|---|---|
| Imbalance Ratio ($\rho$) | 100 | 50 | 10 | 100 | 50 | 10 |
| CE | 29.64 | 25.19 | 13.61 | 61.68 | 56.15 | 44.29 |
| CB-CE [11] | 27.63 | 21.95 | 13.23 | 61.44 | 55.45 | 42.88 |
| CB-CE + *SSP* | **23.47** | **19.60** | **11.57** | **56.94** | **52.91** | **41.94** |
| Focal [32] | 29.62 | 23.29 | 13.34 | 61.59 | 55.68 | 44.22 |
| CB-Focal [11] | 25.43 | 20.73 | 12.90 | 60.40 | 54.83 | 42.01 |
| CB-Focal + *SSP* | **22.90** | **18.74** | **11.75** | **57.03** | **53.12** | **41.16** |
| CE-DRW [7] | 24.94 | 21.10 | 13.57 | 59.49 | 55.31 | 43.78 |
| CE-DRS [7] | 25.53 | 21.39 | 13.73 | 59.62 | 55.46 | 43.95 |
| CE-DRW + *SSP* | **23.04** | **19.93** | **12.66** | **57.21** | **53.57** | **41.77** |
| LDAM [7] | 26.65 | 23.18 | 13.04 | 60.40 | 55.03 | 43.09 |
| LDAM-DRW [7] | 22.97 | 19.06 | 11.84 | 57.96 | 53.85 | 41.29 |
| LDAM-DRW + *SSP* | **22.17** | **17.87** | **11.47** | **56.57** | **52.89** | **41.09** |

## 4.2 Self-Supervised Imbalanced Learning Framework

Motivated by our theoretical results, we again seek to systematically study how self-supervision can help and improve class-imbalanced tasks in realistic settings.

**Self-Supervised Imbalanced Learning.** To utilize self-supervision for overcoming the intrinsic label bias, we propose to, in the first stage of learning, abandon the label information and perform *self-supervised pre-training* (SSP). This procedure aims to learn better initialization that is more label-agnostic from the imbalanced dataset. After the first stage of learning with self-supervision, we can then perform any standard training approach to learn the final model initialized by the pre-trained network. Since the pre-training is independent of the learning approach applied in the normal training stage, such strategy is compatible with any existing imbalanced learning techniques.

Once the self-supervision yields good initialization, the network can benefit from the pre-training tasks and finally learn more generalizable representations. Since SSP can be easily embedded with existing techniques, we would expect that any base classifiers can be consistently improved using SSP. To this end, we empirically evaluate SSP and show that it leads to consistent and substantial improvements in class-imbalanced learning, across various large-scale long-tailed benchmarks.

**Experimental Setup.** We perform extensive experiments on benchmark CIFAR-10-LT and CIFAR-100-LT, as well as large-scale long-tailed datasets including ImageNet-LT [33] and real-world dataset iNaturalist 2018 [24]. We again evaluate models on the corresponding balanced test datasets [7,25,33]. We use Rotation [16] as SSP method on CIFAR-LT, and MoCo [19] on ImageNet-LT and iNaturalist. In the classifier learning stage, we follow [7,25] to train all models for 200 epochs on CIFAR-LT, and 90 epochs on ImageNet-LT and iNaturalist. Other implementation details are in Appendix D.3.

### 4.2.1 Main Results

**CIFAR-10-LT & CIFAR-100-LT.** We present imbalanced classification on long-tailed CIFAR in Table 2. We select standard cross-entropy (CE) loss, Focal loss [32], class-balanced (CB) loss [11], re-weighting or re-sampling training schedule [7], and recently proposed LDAM-DRW [7] as state-of-the-art methods. We group the competing methods into four sessions according to which basic loss or learning strategies they use. As Table 2 reports, in each session across different $\rho$, adding SSP consistently outperforms the competing ones by notable margins. Further, the benefits of SSP become more significant as $\rho$ increases, demonstrating the value of self-supervision under class imbalance.

**ImageNet-LT & iNaturalist 2018.** Besides standard and balanced CE training, we also select other baselines including OLTR [33] and recently proposed classifier re-training (cRT) [25] which achieves state-of-the-art on large-scale datasets. Table 3 and 4 present results on two datasets, respectively.

Table 3: Top-1 test error rates (%) on ImageNet-LT. † denotes results reproduced with authors' code.

| Method | ResNet-10 | ResNet-50 |
|---|---|---|
| CE (Uniform) | 65.2 | 61.6 |
| CE (Uniform) + *SSP* | **64.1** | **54.4** |
| CE (Balanced) | 62.9 | 59.7 |
| CE (Balanced) + *SSP* | **61.6** | **52.4** |
| OLTR [33] | 64.4 | 62.6† |
| OLTR + *SSP* | **62.3** | **53.9** |
| cRT [25] | 58.2 | 52.7 |
| cRT + *SSP* | **56.8** | **48.7** |

Table 4: Top-1 test error rates (%) on iNaturalist 2018. † denotes results reproduced with authors' code.

| Method | ResNet-50 |
|---|---|
| CE (Uniform) | 39.3 |
| CE (Uniform) + *SSP* | **35.6** |
| CE (Balanced) | 36.5 |
| CE (Balanced) + *SSP* | **34.1** |
| LDAM-DRW [7] | 35.4† |
| LDAM-DRW + *SSP* | **33.7** |
| cRT [25] | 34.8 |
| cRT + *SSP* | **31.9** |

On both datasets, adding SSP sets new state-of-the-arts, substantially improving current techniques with 4% absolute performance gains. The consistent results confirm the success of applying SSP in realistic large-scale imbalanced learning scenarios.

**Qualitative Results.** To gain additional insight, we look at the t-SNE projection of learnt representations for both vanilla CE training (Fig. 4a) and with SSP (Fig. 4b). For each method, the projection is performed over both training and test data, thus providing the same decision boundary for better visualization. The figures show that the decision boundary of vanilla CE can be greatly altered by the head classes, which results in the large leakage of tail classes during (balanced) inference. In contrast, using SSP sustains clear separation with less leakage, especially between adjacent head and tail class.

**Summary.** Regardless of the settings and the base training techniques, adding our self-supervision framework in the first stage of learning can uniformly boost the final performance. This highlights that the negative, "unnecessary" viewpoint of the imbalanced labels is also valuable and effective in improving the state-of-the-art imbalanced learning approaches.

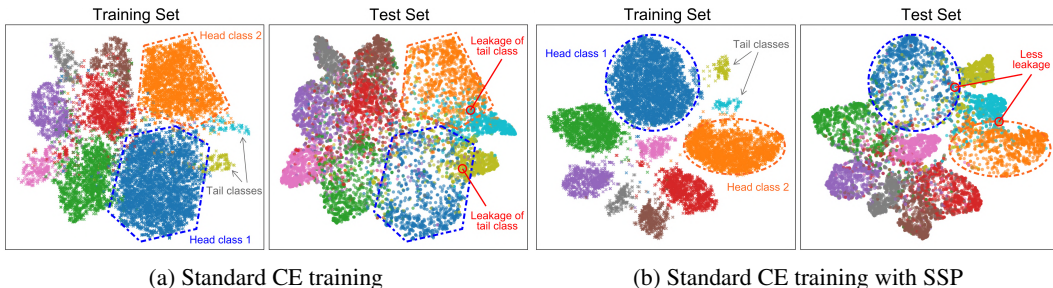

(a) Standard CE training          (b) Standard CE training with SSP

Figure 4: t-SNE visualization of training & test set on CIFAR-10-LT. Using SSP helps mitigate the tail classes leakage during testing, which results in better learned boundaries and representations.

### 4.2.2 Further Analysis and Ablation Studies

**Different Self-Supervised Methods (Appendix F.1).** We select four different SSP techniques, and evaluate them across four benchmark datasets. In general, all SSP methods can lead to notable gains compared to the baseline, while interestingly the gain varies across methods. We find that MoCo [19] performs better on large-scale datasets, while Rotation [16] achieves better results on smaller ones.

**Generalization on Minority Classes (Appendix F.2).** In addition to the top-1 accuracy, we further study the generalization on each specific class. On both CIFAR-10-LT and ImageNet-LT, we observe that SSP can lead to consistent gains across all classes, where trends are more evident for tail classes.

**Imbalance Type (Appendix F.3).** While the main paper is focused on the long-tailed imbalance distribution which is the most common type of imbalance, we remark that other imbalance types are also suggested in literature [5]. We present ablation study on another type of imbalance, i.e., step imbalance [5], where consistent improvements and conclusions are verified when adding SSP.

# 5  Related Work

**Imbalanced Learning & Long-tailed Recognition.** Literature is rich on learning long-tailed imbalanced data. Classical methods have been focusing on designing data re-sampling strategies, including over-sampling the minority classes [2, 41, 44] as well as under-sampling the frequent classes [5, 31]. In addition, cost-sensitive re-weighting schemes [6, 22, 23, 27, 52] are proposed to (dynamically) adjust weights during training for different classes or even different samples. For imbalanced classification problems, another line of work develops class-balanced losses [7, 11, 13, 26, 32] by taking intra- or inter-class properties into account. Other learning paradigms, including transfer learning [33, 54], metric learning [55, 58], and meta-learning [1, 45], have also been explored. Recent studies [25, 59] also find that decoupling the representation and classifier leads to better long-tailed learning results. In contrast to existing works, we provide systematic strategies through two viewpoints of imbalanced labels, which boost imbalanced learning in both semi-supervised and self-supervised manners.

**Semi-Supervised Learning.** Semi-supervised learning is concerned with learning from both unlabeled and labeled samples, where typical methods ranging from entropy minimization [18], pseudo-labeling [30], to generative models [15, 17, 29, 42, 56]. Recently, a line of work that proposes to use consistency-based regularization approaches has shown promising performance in semi-supervised tasks, where consistency loss is integrated to push the decision boundary to low-density areas using unlabeled data [3, 28, 35, 43, 46, 50]. The common evaluation protocol assumes the unlabeled data comes from the same or similar distributions as labeled data, while authors in [39] argue it may not reflect realistic settings. In our work, we consider the data imbalance in both labeled and unlabeled datasets, as well as the data relevance for the unlabeled data, which altogether provides a principled setup on semi-supervised learning for imbalanced learning tasks.

**Self-Supervised Learning.** Learning with self-supervision has recently attracted increasing interests, where early approaches mainly rely on pretext tasks, including exemplar classification [14], predicting relative locations of image patches [12], image colorization [57], solving jigsaw puzzles of image patches [37], object counting [38], clustering [9], and predicting the rotation of images [16]. More recently, a line of work based on contrastive losses [4, 19, 21, 40, 47] shows great success in self-supervised representation learning, where similar embeddings are learned for different views of the same training example, and dissimilar embeddings for different training examples. Our work investigates self-supervised pre-training in class-imbalanced context, revealing surprising yet intriguing findings on how the self-supervision can help alleviate the biased label effect in imbalanced learning.

# 6  Conclusion

We systematically study the value of labels in class-imbalanced learning, and propose two theoretically grounded strategies to understand, validate, and leverage such imbalanced labels in both semi- and self-supervised manners. On both sides, sound theoretical guarantees as well as superior performance on large-scale imbalanced datasets are demonstrated, confirming the significance of the proposed schemes. Our findings open up new avenues for learning imbalanced data, highlighting the need to rethink what is the best way to leverage the inherently biased labels to improve imbalanced learning.

# Broader Impact

Real-world data often exhibits skewed distributions with a long tail, rather than the ideal uniform distributions over each class. We tackle this important problem through two novel perspectives: (1) using unlabeled data without depending on additional human labeling; (2) explore intrinsic properties from data itself with self-supervision. These simple yet effective strategies introduce new frameworks for improving generic imbalanced learning tasks, which we believe will broadly benefit practitioners dealing with heavily imbalanced data in realistic applications.

On the other hand, however, we only extensively test our strategies on academic datasets. In many real-world applications such as autonomous driving, medical diagnosis, and healthcare, beyond being naturally imbalanced, the data may impose additional constraints on learning process and final models, e.g., being fair or private. We focus on standard accuracy as our measure and largely ignore other ethical issues in imbalanced data, especially in minor classes. As such, the risk of producing unfair or biased outputs reminds us to carry rigorous validations in critical, high-stakes applications.

## Acknowledgments and Disclosure of Funding

The authors thank the anonymous reviewers for their insightful comments and helpful discussions. This research did not receive any specific grant from funding agencies in the public, commercial, or not-for-profit sectors.

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
