[Supplementary Material]

# Appendices

## A  Proof of Theorem 1

Given the pseudo-labels, we note that we can rewrite $\tilde{X}_i^+ = (1 - I_i^+)(\mu_2 - \mu_1) + Z_i^+$, where $Z_i^+ \sim \mathcal{N}(\mu_1, \sigma^2)$ and $I_i^+ \sim \text{Bernoulli}(p)$. That is, if the pseudo-label is correct, then $\tilde{X}_i^+ \sim Z_i^+$ and otherwise, $\tilde{X}_i^+ \sim (\mu_2 - \mu_1) + Z_i^+$. Similarly, $\tilde{X}_i^- = (1 - I_i^-)(\mu_1 - \mu_2) + Z_i^-$, where $Z_i^- \sim \mathcal{N}(\mu_2, \sigma^2)$ and $I_i^- \sim \text{Bernoulli}(q)$. Now,

$$
\begin{aligned}
\hat{\theta} &= \frac{1}{2}\left( \frac{\sum_{i=1}^{\tilde{n}_+} \tilde{X}_i^+}{\tilde{n}_+} + \frac{\sum_{i=1}^{\tilde{n}_-} \tilde{X}_i^-}{\tilde{n}_-} \right) \\
&= \frac{1}{2}\left\{ \frac{\sum_{i=1}^{\tilde{n}_+} (1 - I_i^+)(\mu_2 - \mu_1) + Z_i^+}{\tilde{n}_+} + \frac{\sum_{i=1}^{\tilde{n}_-} (1 - I_i^-)(\mu_1 - \mu_2) + Z_i^-}{\tilde{n}_-} \right\} \\
&= \frac{1}{2}\left\{ \frac{\sum_{i=1}^{\tilde{n}_+} I_i^+(\mu_1 - \mu_2)}{\tilde{n}_+} + \frac{\sum_{i=1}^{\tilde{n}_-} I_i^-(\mu_2 - \mu_1)}{\tilde{n}_-} + \left( \frac{\sum_{i=1}^{\tilde{n}_+} Z_i^+}{\tilde{n}_+} + \frac{\sum_{i=1}^{\tilde{n}_-} Z_i^-}{\tilde{n}_-} \right) \right\}.
\end{aligned}
\tag{1}
$$

We bound each term in Eq. (1) separately. First, note that

$$
\left( \frac{\sum_{i=1}^{\tilde{n}_+} Z_i^+}{\tilde{n}_+} + \frac{\sum_{i=1}^{\tilde{n}_-} Z_i^-}{\tilde{n}_-} \right) \sim \mathcal{N}\left( \mu_1 + \mu_2,\ \sigma^2\left( \frac{1}{\tilde{n}_+} + \frac{1}{\tilde{n}_-} \right) \right).
$$

By standard Gaussian concentration, we have

$$
\mathbb{P}\left( \left| \frac{\sum_{i=1}^{\tilde{n}_+} Z_i^+}{\tilde{n}_+} + \frac{\sum_{i=1}^{\tilde{n}_-} Z_i^-}{\tilde{n}_-} - (\mu_1 + \mu_2) \right| > t \right) \le 2e^{-\frac{t^2}{2\sigma^2} \cdot \frac{1}{\frac{1}{\tilde{n}_+} + \frac{1}{\tilde{n}_-}}}.
\tag{2}
$$

Next, we bound the term $\frac{\sum_{i=1}^{\tilde{n}_+} I_i^+}{\tilde{n}_+}$. Since $I_i^+ \sim \text{Bernoulli}(p)$, applying Hoeffding inequality

$$
\mathbb{P}\left( \left| \frac{\sum_{i=1}^{\tilde{n}_+} I_i^+}{\tilde{n}_+} - p \right| > t \right) \le 2e^{-2\tilde{n}_+ t^2}.
\tag{3}
$$

Similarly, we have

$$
\mathbb{P}\left( \left| \frac{\sum_{i=1}^{\tilde{n}_-} I_i^-}{\tilde{n}_-} - q \right| > t \right) \le 2e^{-2\tilde{n}_- t^2}.
\tag{4}
$$

Note that by triangle inequality,

$$
\begin{aligned}
&\left| \hat{\theta} - \frac{\Delta(\mu_1 - \mu_2)}{2} - \frac{(\mu_1 + \mu_2)}{2} \right| \\
&= \left| \frac{1}{2}\left( \frac{\sum_{i=1}^{\tilde{n}_+} \tilde{X}_i^+}{\tilde{n}_+} + \frac{\sum_{i=1}^{\tilde{n}_-} \tilde{X}_i^-}{\tilde{n}_-} \right) - \frac{p(\mu_1 - \mu_2)}{2} - \frac{q(\mu_2 - \mu_1)}{2} - \frac{(\mu_1 + \mu_2)}{2} \right| \\
&\le \left| \frac{1}{2} \frac{\sum_{i=1}^{\tilde{n}_+} I_i^+}{\tilde{n}_+} - \frac{p}{2} \right| |\mu_1 - \mu_2| + \left| \frac{1}{2} \frac{\sum_{i=1}^{\tilde{n}_-} I_i^-}{\tilde{n}_-} - \frac{q}{2} \right| |\mu_1 - \mu_2| \\
&\quad + \left| \frac{1}{2}\left( \frac{\sum_{i=1}^{\tilde{n}_+} Z_i^+}{\tilde{n}_+} + \frac{\sum_{i=1}^{\tilde{n}_-} Z_i^-}{\tilde{n}_-} \right) - \frac{\mu_1 + \mu_2}{2} \right|.
\end{aligned}
$$

Given $\delta > 0$, consider the event that

$$
E = \left\{ \left| \frac{1}{2} \frac{\sum_{i=1}^{\tilde{n}_+} I_i^+}{\tilde{n}_+} - \frac{p}{2} \right| |\mu_1 - \mu_2| \le \frac{\delta}{3} \quad \text{and} \quad \left| \frac{1}{2} \frac{\sum_{i=1}^{\tilde{n}_-} I_i^-}{\tilde{n}_-} - \frac{q}{2} \right| |\mu_1 - \mu_2| \le \frac{\delta}{3} \quad \text{and} \right.
$$

$$
\left. \left| \frac{1}{2}\left( \frac{\sum_{i=1}^{\tilde{n}_+} Z_i^+}{\tilde{n}_+} + \frac{\sum_{i=1}^{\tilde{n}_-} Z_i^-}{\tilde{n}_-} \right) - \frac{\mu_1 + \mu_2}{2} \right| \le \frac{\delta}{3} \right\}.
$$

Using union bound and the concentration inequalities Eqs. (2), (3) and (4), we obtain the following lower bound on the probability of event $E$:

$$\mathbb{P}(E) \geq 1 - 2e^{-\frac{2\delta^2}{9\sigma^2} \cdot \frac{1}{\frac{1}{\tilde{n}_+} + \frac{1}{\tilde{n}_-}}} - 2e^{-\frac{8\tilde{n}_+ \delta^2}{9(\mu_1 - \mu_2)^2}} - 2e^{-\frac{8\tilde{n}_- \delta^2}{9(\mu_1 - \mu_2)^2}}.$$

Finally, the above equation and the triangle inequality implies that

$$\mathbb{P}\left( \left| \hat{\theta} - \frac{\Delta(\mu_1 - \mu_2)}{2} - \frac{(\mu_1 + \mu_2)}{2} \right| \leq \delta \right) \geq 1 - 2e^{-\frac{2\delta^2}{9\sigma^2} \cdot \frac{1}{\frac{1}{\tilde{n}_+} + \frac{1}{\tilde{n}_-}}} - 2e^{-\frac{8\tilde{n}_+ \delta^2}{9(\mu_1 - \mu_2)^2}} - 2e^{-\frac{8\tilde{n}_- \delta^2}{9(\mu_1 - \mu_2)^2}}.$$

This completes the proof.

## B  Proof of Theorem 2

Note that if $X \sim \mathcal{N}(0, \sigma^2 \mathbf{I}_d)$, then for a given $\theta$, $\theta^T X \sim \mathcal{N}(0, ||\theta||_2^2 \sigma^2)$. Based on the form of the linear classifier, we know that

$$
\begin{aligned}
\mathrm{err}_f &= \mathbb{P}_{(X,Y) \sim P_{XY}} \Big( y(\theta^T X + b) < 0 \Big) \\
&= \mathbb{P}_{(X,Y) \sim P_{XY}} \Big( y(\theta^T X + b) < 0 \big| Y = +1 \Big) \mathbb{P}\Big( Y = +1 \Big) \\
&\quad + \mathbb{P}_{(X,Y) \sim P_{XY}} \Big( y(\theta^T X + b) < 0 \big| Y = -1 \Big) \mathbb{P}\Big( Y = -1 \Big) \\
&= p_+ \mathbb{P}\Big( \mathcal{N}(0, ||\theta||_2^2 \sigma_1^2) < -b \Big) + p_- \mathbb{P}\Big( \mathcal{N}(0, ||\theta||_2^2 \beta \sigma_1^2) > -b \Big) \\
&= p_+ \mathbb{P}\left( \mathcal{N}(0,1) < -\frac{b}{||\theta||_2 \sigma_1} \right) + p_- \mathbb{P}\left( \mathcal{N}(0,1) < \frac{b}{||\theta||_2 \sqrt{\beta} \sigma_1} \right) \\
&= p_+ \Phi\left( -\frac{b}{||\theta||_2 \sigma_1} \right) + p_- \Phi\left( \frac{b}{||\theta||_2 \sqrt{\beta} \sigma_1} \right).
\end{aligned}
$$

Finally, note that for $b > 0$, $\Phi\left( \frac{b}{||\theta||_2 \sqrt{\beta} \sigma_1} \right) \geq 1/2$. Since $p_-$ is assumed to be at least $1/2$, the error probability of the classifier is at least $1/4$.

## C  Proof of Theorem 3

We recall the following standard concentration inequality for sub-exponential random variables [51]. Suppose that $W_1, W_2, \ldots, W_n$ are i.i.d. sub-exponential random variables with parameters $(\nu, \alpha)$. Then,

$$\mathbb{P}\left( \frac{1}{n} \sum_{i=1}^{n} (W_i - E[X_i]) \geq \delta \right) \leq \begin{cases} e^{-\frac{n\delta^2}{2\nu^2}}, & \text{for } 0 \leq \delta \leq \frac{\nu^2}{\alpha}; \\ e^{-\frac{n\delta}{2\alpha}}, & \text{for } \delta > \frac{\nu^2}{\alpha}. \end{cases}$$

We remark that a similar two-sided tail bounds also hold. For our purpose, note that if $W \sim \mathcal{N}(0,1)$, then $W^2$ is sub-exponential with parameter $(2,4)$. Therefore, we have the standard $\chi^2$-concentration for a $\chi^2$ random variable with degree $n$ as follows:

$$\mathbb{P}\left( \left| \frac{1}{n} \sum_{i=1}^{n} W_i^2 - 1 \right| \geq \delta \right) \leq 2e^{-n\delta^2/8}, \quad \forall \delta \in (0,1).$$

Let us now analyze the classifier $f_{ss}(X) = \mathrm{sign}(-Z + b)$ obtained by self-supervised training. Recall that we assume $Z = \psi(X) = k_1 ||X||_2^2 + k_2$. For the negative training data, we note that $\frac{Z_i - k_2}{k_1 \beta \sigma_1^2} \sim \chi_d^2$ (the $\chi^2$ distribution with $d$ degree of freedom). Using the previous $\chi^2$ concentration, we have that for $\delta \in (0,1)$,

$$\mathbb{P}\left( \left| \frac{1}{N_- d} \sum_{i=1}^{N} \mathbf{1}_{\{Y_i = -1\}} \frac{Z_i - k_2}{k_1 \beta \sigma_1^2} - 1 \right| \geq \delta \right) \leq 2e^{-N_- d\delta^2/8}.$$

Rearrange the terms, we obtain

$$\mathbb{P}\left(\left|\frac{1}{N_-}\sum_{i=1}^{N}\mathbf{1}_{\{Y_i=-1\}}Z_i - \left(dk_1\beta\sigma_1^2 + k_2\right)\right| \geq \delta dk_1\beta\sigma_1^2\right) \leq 2e^{-N_-d\delta^2/8}.$$

Similarly, for the positive training data, we have

$$\mathbb{P}\left(\left|\frac{1}{N_+}\sum_{i=1}^{N}\mathbf{1}_{\{Y_i=+1\}}Z_i - \left(dk_1\sigma_1^2 + k_2\right)\right| \geq \delta dk_1\sigma_1^2\right) \leq 2e^{-N_+d\delta^2/8}.$$

Therefore, by an application of union bound, with probability at least $1 - 2e^{-N_-d\delta^2/8} - 2e^{-N_+d\delta^2/8}$, the intercept $b$ of the self-supervised classifier, i.e., $b = \frac{1}{2}\left(\frac{\sum_{i=1}^{N}\mathbf{1}_{\{Y_i=+1\}}Z_i}{N_+} + \frac{\sum_{i=1}^{N}\mathbf{1}_{\{Y_i=-1\}}Z_i}{N_-}\right) > 0$, satisfies

$$\left|b - \frac{dk_1(\beta+1)\sigma_1^2}{2} - k_2\right| \leq \frac{\delta dk_1(\beta+1)\sigma_1^2}{2}. \tag{5}$$

In the following, we condition on the event that $b$ satisfies the bound in Eq. (5). Then,

$$\begin{aligned}
\mathrm{err}_{f_{ss}} &= \mathbb{P}_{(X,Y)\sim P_{XY}}\left(y(-Z+b) < 0\right) \\
&= \mathbb{P}_{(X,Y)\sim P_{XY}}\left(y(-Z+b) < 0 | Y = +1\right)\mathbb{P}\left(Y = +1\right) \\
&\quad + \mathbb{P}_{(X,Y)\sim P_{XY}}\left(y(-Z+b) < 0 | Y = -1\right)\mathbb{P}\left(Y = -1\right) \\
&= p_+\mathbb{P}_{(X,Y)\sim P_{XY}}\left(Z > b | Y = +1\right) + p_-\mathbb{P}_{(X,Y)\sim P_{XY}}\left(Z < b | Y = -1\right) \\
&\leq p_+\mathbb{P}_{(X,Y)\sim P_{XY}}\left(Z > \frac{dk_1(\beta+1)\sigma_1^2}{2} + k_2 - \frac{\delta dk_1(\beta+1)\sigma_1^2}{2}\bigg| Y = +1\right) \\
&\quad + p_-\mathbb{P}_{(X,Y)\sim P_{XY}}\left(Z < \frac{dk_1(\beta+1)\sigma_1^2}{2} + k_2 + \frac{\delta dk_1(\beta+1)\sigma_1^2}{2}\bigg| Y = -1\right). \tag{6}
\end{aligned}$$

We analyze the two terms in the bound Eq. (6). First, condition on $Y = -1$, again note that $\frac{Z-k_2}{k_1\beta\sigma_1^2} \sim \chi_d^2$. Therefore,

$$\begin{aligned}
&\mathbb{P}\left(Z < \frac{dk_1(\beta+1)\sigma_1^2}{2} + k_2 + \frac{\delta dk_1(\beta+1)\sigma_1^2}{2}\bigg| Y = -1\right) \\
&= \mathbb{P}\left(Z - (dk_1\beta\sigma_1^2 + k_2) < -\frac{(\beta-1-\delta(\beta+1))}{2\beta}dk_1\beta\sigma_1^2\bigg| Y = -1\right) \\
&\leq \exp\left(-d \cdot \frac{(\beta-1-\delta(\beta+1))^2}{32\beta^2}\right),
\end{aligned}$$

where the last inequality follows from the concentration inequality and note that $0 < (\beta - 1 - \delta(\beta+1))/2\beta < 1$. In a similar manner, for $Y = +1$, note that $\frac{Z-k_2}{k_1\sigma_1^2} \sim \chi_d^2$. Hence,

$$\begin{aligned}
&\mathbb{P}\left(Z > \frac{dk_1(\beta+1)\sigma_1^2}{2} + k_2 - \frac{\delta dk_1(\beta+1)\sigma_1^2}{2}\bigg| Y = +1\right) \\
&= \mathbb{P}\left(Z - (dk_1\sigma_1^2 + k_2) > \frac{(\beta-1-\delta(\beta+1))}{2}dk_1\sigma_1^2\bigg| Y = +1\right) \\
&\leq \begin{cases} \exp\left(-d \cdot \frac{(\beta-1-\delta(\beta+1))^2}{32}\right), & \text{if } \delta \in \left[\frac{\beta-3}{\beta+1}, \frac{\beta-1}{\beta+1}\right); \\ \exp\left(-d \cdot \frac{(\beta-1-\delta(\beta+1))}{16}\right), & \text{if } \delta \in \left(0, \frac{\beta-3}{\beta+1}\right). \end{cases}
\end{aligned}$$

For the last inequality, we note that if $\delta \in \left[\frac{\beta-3}{\beta+1}, \frac{\beta-1}{\beta+1}\right)$, $0 < \frac{(\beta-1-\delta(\beta+1))}{2} \leq 1$; otherwise, $\frac{(\beta-1-\delta(\beta+1))}{2} > 1$. Substituting the above inequalities into the error probability $\mathrm{err}_{f_{ss}}$ (Eq. (6)) completes the proof.

# D Experimental Details

## D.1 Imbalanced Dataset Details

In this section, we provide the detailed information of five long-tailed imbalanced datasets we use in our experiments. Table 5 provides an overview of the five datasets.

Table 5: Overview of the five imbalanced datasets used in our experiments. $\rho$ indicates the imbalance ratio.

| Dataset | # Class | $\rho$ | Head class size | Tail class size | # Training set | # Val. set | # Test set |
|---|---|---|---|---|---|---|---|
| CIFAR-10-LT | 10 | $10 \sim 100$ | 5,000 | $500 \sim 50$ | $20{,}431 \sim 12{,}406$ | — | 10,000 |
| CIFAR-100-LT | 100 | $10 \sim 100$ | 500 | $50 \sim 5$ | $19{,}573 \sim 10{,}847$ | — | 10,000 |
| SVHN-LT | 10 | $10 \sim 100$ | 1,000 | $100 \sim 10$ | $4{,}084 \sim 2{,}478$ | — | 26,032 |
| ImageNet-LT | 1,000 | 256 | 1,280 | 5 | 115,846 | 20,000 | 50,000 |
| iNaturalist 2018 | 8,142 | 500 | 1,000 | 2 | 437,513 | 24,426 | — |

**CIFAR-10-LT and CIFAR-100-LT.** The original versions of CIFAR-10 and CIFAR-100 contain 50,000 images for training and 10,000 images for testing with class number of 10 and 100, respectively. We create the long-tailed CIFAR versions following [7,11] with controllable degrees of data imbalance, and keep the test set unchanged. We vary the class imbalance ratio $\rho$ from 10 to 100.

**SVHN-LT.** The original SVHN dataset contains 73,257 images for training and 26,032 images for testing with 10 classes. Similarly to CIFAR-LT, we create SVHN-LT dataset with maximally 1,000 images per class (head class), and vary $\rho$ from 10 to 100 for different long-tailed versions.

**ImageNet-LT.** ImageNet-LT [33] is artificially truncated from its balanced version, with sample size in the training set following an exponential decay across different classes. ImageNet-LT has 1,000 classes and 115.8K training images, with number of images ranging from 1,280 to 5 images per class.

**iNaturalist 2018.** iNaturalist 2018 [24] is a real-world fine-grained visual recognition dataset that naturally exhibits long-tailed class distributions, consisting of 435,713 samples from 8,142 species.

## D.2 Unlabeled Data Details

We provide additional information on the unlabeled data we use in the semi-supervised settings, i.e., the unlabeled data sourcing, and how we create unlabeled sets with different imbalanced distributions.

**Unlabeled Data Sourcing.** To obtain the unlabeled data needed for our semi-supervised setup, we follow [8] to mine the 80 Million Tiny Images (80M) dataset [48] to source unlabeled and uncurated data for CIFAR-10. In particular, CIFAR-10 is a human-labeled subset of 80M, which is manually restricted to 10 classes. Accordingly, most images in 80M do not belong to any image categories in CIFAR-10. To select unlabeled data that exhibit similar distributions as labeled ones, we follow the procedure as in [8], where an 11-class classification model is trained to distinguish CIFAR-10 classes and an "non-CIFAR" class. For each class, we then rank the images based on the prediction confidence, and construct the unlabeled (imbalanced) dataset $\mathcal{D}_U$ according to our settings.

For SVHN, since its own dataset contains an extra part [36] with 531.1K additional (labeled) samples, we directly use these additional data to simulate the unlabeled dataset, which exhibits similar data distribution as the main dataset. Specifically, the ground truth labels are used only for preparing $\mathcal{D}_U$, and are abandoned throughout experiments (i.e., before performing pseudo-labeling).

**Relevant (and Irrelevant) Unlabeled Data.** To analyze the data relevance of unlabeled data with class imbalance (cf. Sec. 3), we again employ the 11-way classifier to select samples with prediction scores that are high for the extra class, and use them as proxy for irrelevant data. We then mix the irrelevant dataset and our main unlabeled dataset with different proportions, thus creating a sequence of unlabeled datasets with different degrees of data relevance.

**Unlabeled Data with Class Imbalance.** With the sourced unlabeled data, we construct the demanded unlabeled dataset $\mathcal{D}_U$ also with class imbalance. In Fig. 5, we show an example of data distributions of both original labeled imbalanced dataset $\mathcal{D}_L$ and $\mathcal{D}_U$ with different unlabeled imbalance ratio. Specifically, Fig. 5a presents the training and test set of CIFAR-10-LT with $\rho = 100$, where a long

Figure 5: An illustration of labeled dataset ($\mathcal{D}_L$) as well as its corresponding unlabeled dataset ($\mathcal{D}_U$ @5x) under different unlabeled imbalance ratio $\rho_U$. Given $\mathcal{D}_L$ with a fixed $\rho$, the total amount of $\mathcal{D}_U$ is fixed, while different $\rho_U$ will lead to different class distributions of the unlabeled data.

tail can be observed for the training set, while the test set is balanced across classes. With labeled data on hand, we create different degrees of class imbalance in $\mathcal{D}_U$ to be (1) uniform ($\rho_U = 1$, Fig. 5b), (2) half imbalanced as labeled set ($\rho_U = \rho/2$, Fig. 5c), (3) same imbalanced ($\rho_U = \rho$, Fig. 5d), and (4) double imbalanced ($\rho_U = 2\rho$, Fig. 5e). Note that the total data amount of $\mathcal{D}_U$ is fixed (e.g., 5x as labeled set) given $\mathcal{D}_L$, while different $\rho_U$ will result in different unlabeled data distributions.

### D.3 Implementation Details

**CIFAR-10-LT and CIFAR-100-LT.** Following [1,7,11], we use ResNet-32 [20] for all CIFAR-LT experiments. The data augmentation follows [20] to use zero-padding with 4 pixels on each side and then random crop back to the original image size, after which a random horizontal flip is performed. We train all models for 200 epochs, and remain all other hyper-parameters the same as [7]. In the semi-supervised settings, we fix the unlabeled weight $\omega = 1$ for all experiments.

**SVHN-LT.** Similarly to CIFAR-LT, we use ResNet-32 model for all SVHN-LT experiments, and fix the same hyper-parameters as in CIFAR-LT experiments throughout training.

**ImageNet-LT.** We follow [25,33] to report results with ResNet-10 and ResNet-50 models. Since [33] only employs ResNet-10 model, we reproduce the results with ResNet-50 using the public code from the authors for fair comparison. During the classifier training stage, we train all models for 90 epochs, and keep all other hyper-parameters identical to those in [25]. During the self-supervised pre-training stage, we leave the hyper-parameters unchanged as in [19], but only use samples from ImageNet-LT.

**iNaturalist 2018.** We follow [1,7,11,25] to use ResNet-50 model. Similar to ImageNet-LT, we train all models for 90 epochs in the classifier training stage, and other hyper-parameters are kept the same as in [25]. The self-supervised pre-training stage is remained the same as that on ImageNet-LT. We reproduce the results for [7] on iNaturalist 2018 using the authors' code.

## E   Additional Results for Semi-Supervised Imbalanced Learning

### E.1   Different Semi-Supervised Learning Methods

We study the effect of different advanced semi-supervised learning methods, in addition to the simple pseudo-label strategy we apply in the main text. We select the following two methods for analysis.

**Virtual Adversarial Training.** Virtual adversarial training (VAT) [35] is one of the state-of-the-art semi-supervised learning methods, which aims to make the predicted labels robust around input data point against local perturbation. It approximates a tiny perturbation $\epsilon_{adv}$ to add to the (unlabeled) inputs which would most significantly affect the outputs of the model. Note that the implementation difference between VAT and the pseudo-label is the loss term on the unlabeled data, where VAT exhibits a consistency regularization loss rather than supervised loss, resulting in a loss function as $\mathcal{L}(\mathcal{D}_L, \theta) + \omega \mathcal{L}_{\text{con}}(\mathcal{D}_U, \theta)$. We add an additional entropy regularization term following [35].

**Mean Teacher.** The mean teacher (MT) [46] method is also a representative algorithm using consistency regularization, where a teacher model and a student model are maintained and a consistency cost between the student's and the teacher's outputs is introduced. The teacher weights are updated through an exponential moving average (EMA) of the student weights.

Similar to pseudo-label, the two semi-supervised methods can be seamlessly incorporated with our imbalanced learning framework. We present the results with these methods in Table 6. For each run, we construct $\mathcal{D}_U$ @5x with the same imbalance ratio as the labeled set (i.e., $\rho_U = \rho$). As

Table 6: Ablation study of different semi-supervised learning methods on CIFAR-10-LT and SVHN-LT. We fix $\rho_U = \rho$ for each specific setting. Best results of each column are in **bold** and the second best are underlined.

| Dataset | CIFAR-10-LT | | | SVHN-LT | | |
|---|---|---|---|---|---|---|
| Imbalance Ratio ($\rho$) | 100 | 50 | 10 | 100 | 50 | 10 |
| Vanilla CE | 29.64 | 25.19 | 13.61 | 19.98 | 17.50 | 11.46 |
| $\mathcal{D}_U$@5x + Pseudo-label [30] | 18.74 | 18.36 | 10.86 | 14.65 | 13.16 | 10.06 |
| $\mathcal{D}_U$@5x + VAT [35] | 17.93 | 16.53 | **9.44** | 13.07 | 12.27 | 9.29 |
| $\mathcal{D}_U$@5x + MT [46] | **16.52** | **15.79** | 9.53 | **12.34** | **11.12** | **8.62** |

Table 6 reports, across different datasets and imbalance ratios, adding unlabeled data can consistently benefit imbalanced learning via semi-supervised learning. Moreover, by using more advanced SSL techniques, larger improvements can be obtained in general.

### E.2    Class-wise Generalization Results

In the main paper, we report the top-1 test errors as the final performance metric. To gain additional insights on how unlabeled data helps imbalanced tasks, we further look at the generalization results in each class, especially on the minority (tail) classes.

**Generalization on Minority Classes.** In Fig. 6 we plot the test error on each class on CIFAR-10-LT and SVHN-LT with $\rho = 50$. As the figure shows, regardless of the base training technique, using unlabeled data can consistently and substantially improve the generalization on tail classes.

(a) CIFAR-10-LT

(b) SVHN-LT

Figure 6: Class-wise top-1 error rates. C0 stands for the head class, and C9 stands for the tail class. Using unlabeled data leads to better generalization on tail classes while keeping the performance on head classes almost unaffected, and can consistently boost different training techniques. Results are averaged across 5 runs.

**Confusion Matrix.** We further show the confusion matrices on CIFAR-10-LT with and without $\mathcal{D}_U$. Fig. 7 presents the results, where for the vanilla CE training, predictions for tail classes are biased towards the head classes significantly. In contrast, by using unlabeled data, the leakage from tail classes to head classes can be largely eliminated.

(a) Standard CE training

(b) CE with unlabeled data $\mathcal{D}_U$@5x

Figure 7: Confusion matrices of standard CE training and using $\mathcal{D}_U$@5x on CIFAR-10-LT with $\rho = 100$.

## E.3 Effect of Unlabeled Data Amount

We study the effect of the size of the unlabeled dataset on our SSL approach in imbalanced learning. We first fix the labeled dataset $\mathcal{D}_L$ with $\rho = 50$, the unlabeled imbalance ratio to be $\rho_U = \rho$, and then vary the amount of $\mathcal{D}_U$ to be {0.5x,1x,5x,10x} of the size of $\mathcal{D}_L$. Table 7 reports the results, where we can observe that larger $\mathcal{D}_U$ consistently leads to higher gains. Furthermore, even with only 0.5x more unlabeled data, the performance can be boosted largely compared to that without unlabeled data. Interestingly however, as the size of $\mathcal{D}_U$ becomes larger, the gains gradually diminish.

Table 7: Ablation study of how unlabeled data amount affects SSL in imbalanced learning. We fix the imbalance ratios as $\rho = \rho_U = 50$. We vary the amount of $\mathcal{D}_U$ with respect to labeled data amount (e.g., 0.5x means the size of $\mathcal{D}_U$ is half of $\mathcal{D}_L$). Best results of each part are in **bold** and the second best are underlined.

| Dataset | CIFAR-10-LT | | | | SVHN-LT | | | |
|---|---|---|---|---|---|---|---|---|
| $\mathcal{D}_U$ Size (w.r.t. $\mathcal{D}_L$) | 0.5x | 1x | 5x | 10x | 0.5x | 1x | 5x | 10x |
| CE | 25.19 | | | | 17.50 | | | |
| CE + $\mathcal{D}_U$ | 21.75 | 20.35 | 18.36 | **16.88** | 14.96 | 14.13 | 13.16 | **13.02** |
| LDAM-DRW [7] | 19.06 | | | | 14.59 | | | |
| LDAM-DRW + $\mathcal{D}_U$ | 17.43 | 16.59 | 14.93 | **13.91** | 13.93 | 13.07 | 11.26 | **11.09** |

## E.4 Effect of Labeled Data Amount

Following [39], we further study how the labeled data amount affects SSL in imbalanced learning. We fix the imbalance ratios of $\mathcal{D}_L$ and $\mathcal{D}_U$ as $\rho = \rho_U = 50$, and also fix the size of $\mathcal{D}_U$ to be 5x of $\mathcal{D}_L$. We vary $\mathcal{D}_L$ amount to be {0.5x,0.75x,1x} with respect to the original labeled data amount. As Table 8 shows, with smaller size of labeled data, the test errors of vanilla CE training increases largely, while adding unlabeled data can maintain sufficiently low errors. Interestingly, when unlabeled data is added, using only 50% of labeled data can already surpass the fully-supervised baseline on both datasets, demonstrating the power of unlabeled data in the context of imbalanced learning.

Table 8: Ablation study of how labeled data amount affects SSL in imbalanced learning. We fix the imbalance ratios as $\rho = \rho_U = 50$, and fix unlabeled data amount to be 5x of labeled data used. We vary the amount of $\mathcal{D}_L$ with respect to their original labeled data amount (e.g., 0.5x means only half of the initial labeled data is used).

| Dataset | CIFAR-10-LT | | | SVHN-LT | | |
|---|---|---|---|---|---|---|
| $\mathcal{D}_L$ Size | 0.5x | 0.75x | 1x | 0.5x | 0.75x | 1x |
| CE | 33.35 | 28.65 | 25.19 | 23.19 | 19.73 | 17.50 |
| CE + $\mathcal{D}_U$ @5x | 20.77 | 18.67 | **18.36** | 14.80 | 13.51 | **13.16** |

# F Additional Results for Self-Supervised Imbalanced Learning

## F.1 Different Self-Supervised Pre-Training Methods

In this section, we investigate the effect of different SSP methods on imbalanced learning tasks. We select four different SSP approaches, ranging from pretext tasks to recent contrastive methods.

**Solving Jigsaw Puzzles.** Jigsaw [37] is a classical method based on pretext tasks, where an image is divided into patches, and a classifier is trained to predict the correct permutation of these patches.

**Rotation Prediction.** Predicting rotation [16] is another simple yet effective method, where an image is rotated by a random multiple of 90 degrees, constructing a 4-way classification problem; a classifier is then trained to determine the degree of rotation applied to an input image.

**Selfie.** Selfie [49] works by masking out select patches in an image, and then constructs a classification problem to determine the correct patch to be filled in the masked location.

**Momentum Contrast.** The momentum contrast (MoCo) [19] method is one of the recently proposed contrastive techniques, where contrastive losses [19] are applied in a representation space to measure the similarities of positive and negative sample pairs, and a momentum-updated encoder is employed.

We conduct controlled experiments over four benchmark imbalanced datasets, and report the results in Table 9. As the table reveals, all self-supervised pre-training methods can benefit the imbalanced learning, consistently across different datasets. Interestingly however, the performance gain varies across SSP techniques. Specifically, on datasets with smaller scale, i.e., CIFAR-10-LT and CIFAR-100-LT, methods using pretext tasks are generally better than using contrastive learning, with *Rotation* performs the best. In contrast, on larger datasets, i.e., ImageNet-LT and iNaturalist, *MoCo* outperforms other SSP methods by a notable margin. We hypothesize that since MoCo needs a large number of (negative) samples to be effective, the smaller yet imbalanced datasets thus may not benefit much from MoCo, compared to those with larger size and more samples.

Table 9: Ablation study of different self-supervised pre-training methods. We set imbalance ratio of $\rho = 50$ for CIFAR-LT. Best results of each column are in **bold** and the second best are underlined.

| Dataset | CIFAR-10-LT | CIFAR-100-LT | ImageNet-LT | iNaturalist 2018 |
|---|---|---|---|---|
| Vanilla CE | 25.19 | 56.15 | 61.6 | 39.3 |
| + Jigsaw [37] | 24.68 | 55.89 | 60.2 | 38.2 |
| + Selfie [49] | 22.75 | 55.31 | 58.3 | 36.9 |
| + Rotation [16] | **21.80** | **54.96** | 55.7 | 36.5 |
| + MoCo [19] | 24.18 | 55.83 | **54.4** | **35.6** |

## F.2 Class-wise Generalization Results

Similar to the semi-supervised setting, we again take a closer look at generalization on each class to gain further insights, in addition to the top-1 test error rates reported in the main text.

**Generalization on Minority Classes.** We plot the class-wise top-1 errors on CIFAR-10-LT (Fig. 8a) and ImageNet-LT (Fig. 8b), respectively. For ImageNet-LT, we follow [25, 33] to split the test set into three subsets for evaluating shot-wise accuracy, namely *Many-shot* (classes with more than 100 images), *Medium-shot* ($20 \sim 100$ images), and *Few-shot* (less than 20 images). On both datasets, we can observe that regardless of training techniques for the base classifier, using SSP can consistently and substantially improve the generalization on tail (few-shot) classes, while maintaining or slightly improving the performance on head (many-shot) classes. The consistent gains demonstrate the effectiveness of SSP in the context of imbalanced learning, especially for the tail classes.

(a) CIFAR-10-LT

(b) ImageNet-LT

Figure 8: Class-wise top-1 error rates on CIFAR-10-LT and ImageNet-LT. C0 stands for the head class, and C9 stands for the tail class. On ImageNet-LT we follow [33] to report test error on three splits of the set of classes: Many-shot, Medium-shot, and Few-shot. Using SSP leads to better generalization on both head and tail classes, and can consistently boost different training techniques. Results are averaged across 5 runs.

**Confusion Matrix.** To further understand how self-supervision helps imbalanced learning, we again plot the confusion matrices on CIFAR-10-LT with and without SSP, respectively. As illustrated in Fig. 9, the prediction results of vanilla CE training suffers from the large leakage from tail classes to head classes, leading to low accuracy on minority categories. In contrast, by using self-supervised pre-training, the tail-to-head class leakages are greatly compensated, resulting in better performance and consistent improvements across the tail classes.

(a) Standard CE training          (b) Standard CE training with SSP

Figure 9: Confusion matrices of standard CE training and using SSP on CIFAR-10-LT with $\rho = 100$.

### F.3 Effect of Imbalance Type

Finally, we conduct ablation study on another type of imbalance. The majority of the literature [1, 11, 25, 33, 59] focused on the long-tailed imbalance distribution, which is also the typical scenario for large-scale real-world datasets [24, 33]. Yet, few other manually designed imbalance types are also investigated by researchers [5, 7] to provide a comprehensive picture. For completeness, we study the performance of SSP under another imbalance type, i.e., the step imbalance [5], where the training instances of half of the classes (i.e., the minority classes) are reduced to a fixed size. The minority classes are defined to have the same sample size, and so do all frequent classes. The imbalance ratio $\rho$ is the same as in long-tailed setting, i.e., $\rho = \max_i\{n_i\} / \min_i\{n_i\}$.

Table 10 presents the results, where we confirm that SSP can bring in consistent benefits across different imbalanced learning techniques on various datasets. Furthermore, when the dataset is more imbalanced (i.e., with higher $\rho$), the performance gains from SSP tend to be even larger, demonstrating the value of self-supervision under extreme class imbalance.

Table 10: Top-1 test errors (%) of ResNet-32 on CIFAR-10 and CIFAR-100 with step imbalance [5]. Using SSP, we can consistently and substantially improve different imbalanced learning techniques across various datasets, and achieve the best performance. [†] denotes results that reported in [7].

| Dataset | Imbalanced CIFAR-10 | | Imbalanced CIFAR-100 | |
|---|---|---|---|---|
| Imbalance Ratio ($\rho$) | 100 | 10 | 100 | 10 |
| CE | 36.70 | 17.50 | 61.45 | 45.37 |
| CB-CE [11][†] | 38.06 | 16.20 | 78.69 | 47.52 |
| CE + *SSP* | **27.27** | **12.04** | **55.57** | **42.90** |
| Focal [32] | 36.09 | 16.36 | 61.43 | 46.54 |
| CB-Focal [11][†] | 39.73 | 16.54 | 80.24 | 49.98 |
| Focal + *SSP* | **27.00** | **12.07** | **55.12** | **42.93** |
| LDAM [7][†] | 33.42 | 15.00 | 60.42 | 43.73 |
| LDAM-DRW [7][†] | 23.08 | 12.19 | 54.64 | 40.54 |
| LDAM-DRW + *SSP* | **22.95** | **11.83** | **54.28** | **40.33** |