[Reviews · NeurIPS 2020]

Review 1

Summary and Contributions: The paper examines the usefulness of labels in an imbalanced setting. The authors reflect on the positive and negative aspects of using labels, notably when engaging in SSL and self-supervision.

Strengths: This topic is of much interest to the NeurIPS community. The paper combines theoretical justifcations and empirical evaluations.

Weaknesses: It is possible that self-supervision can re-inforce errors. The authors thus have to reflect on using this SSL learning algorithm in their paper. (I understand that other algorithms are used in the Appendix, but this is still the pillar of their contribution.)

Correctness: The authors compare their work to one other algorithm and ignore many other significant contributions in this area. As such, it is difficult to assess the full impact of the contribution.

Clarity: The paper is generally well-written, but the style is very condense, making it sometimes difficult to navigate.

Relation to Prior Work: This is a drawback of the paper; related work are generally ignored or skipped over, even though the list of references is very long.

Reproducibility: Yes

Additional Feedback: The authors are urged to better position their work w.r.t. the substantial works already done in the SSL commuinity.


Review 2

Summary and Contributions: This is a very interesting paper. The contribution of the paper does not rely on proposing a specific semi-supervised and/or self-supervised learning method. Instead, this paper systematically studies both semi-supervised learning (pseudo-labeling) and self-supervised learning in the scenario of class-imbalanced learning, theoretically and empirically. In theory side, the authors propose a few theorems under some simplified conditions to motivate why pseudo-labeling and self-supervised learning can help the class-imbalanced learning. Theorem 1 demonstrates that, in binary classification scenario, pseudo-labeling on more unlabeled data would definitely help; Theorem 3 says, self-supervised learning can improve linear classifier given more data. In term of empirical verification, the authors run quite a lot experiments, in different scenarios and also tried a few self-supervised learning algorithms. The experimental results aligned with the proposed theorems and common sense.

Strengths: The proposed theorems, though under simplified conditions, are very motivating and intuitively explain why data matters for class-imbalanced learning. For experiments, the quantitative evaluation and ablation study all supports the proposed theorems. The results using different loss, different self-supervised learning algorithms, across different data sets are overall consistent. The t-sne plots (figure 1 and 4) are very impressive.

Weaknesses: 1. The authors did not clearly define 'relevance score'. 2. The proposed theorems, do not address the 'relevance score' effect for either semi-supervised or self-supervised learning scenarios. 3. I guess section 3 is largely used to motivate section 4, but I am not sure if you have experiments showing that self-supervised learning is more resistant to 'non-relevant unlabeled data'. I guess self-supervised learning would still be affected if the unlabeled data is totally non-relevant. Like using sounds to improve speech recognition. 4. Title seems to be a little bit misleading, actually, in the SSL scenarios, pseudo labels (and also relevance score) matters; While in the self-supervised learning scenario, the amount of unlabeled data matters. 5. Though very motivating, the theorems need to be further developed to explain more complex models.

Correctness: Yes, the claims be validated by theorems and experiments.

Clarity: Yes, the paper is self-motivating and well-written

Relation to Prior Work: The paper cited recent works in semi-supervised learning, self-supervised learning and class-imbalance learning. There are also some recent semi-supervised learning papers the author can consider to cite, like 'Unsupervised Data Augmentation for Consistency Training'

Reproducibility: Yes

Additional Feedback: Read the authors' and other reviewers' feedback/comments. Intend to keep my evaluation unchanged.


Review 3

Summary and Contributions: This paper provides extensive analysis on how to utilize data and labels in class-imbalanced classification. Through theoretical analysis and empirical evaluations, the authors demonstrate the benefits and potentials of employing semi-supervised learning and self-supervised learning in class-imbalanced classification.

Strengths: 1. The paper focus on analyzing an interesting problem: class-imbalanced classification. The problem is related to commonly existing long-tail issues in many machine learning tasks. The paper provides insightful comments on the effect of available labels in class-imbalanced learning from two different aspects. The results could be of interest to even broader area of different applications. 2. The theoretical analysis is sound. Different factors are considered, such as the class distribution (imbalanceness) and the relevance between training and testing data. Their effects on the learnability and estimation accuracy are both analyzed. 3. Inspired by the theoretical analyzing results, the authors propose to employ pseduo-label strategy to enhance the classification accuracy by harnessing the unlabeled data. Furthermore, self-supervised learning techniques are applied to alleviate the class bias introduced by the imbalanced class distribution. Both methods are proved to be effective through empirical evaluations. 4. The experiments are thorough. Extensive results demonstrate the authors claims clearly.

Weaknesses: Since the paper focuses on the value of labels, it will be stronger if more analysis on effect of different label distributions under class-imbalanced setup. For example, for the long-tail classes, what label distribution is preferable, i.e., should they have same proportion as the data distribution, or they are better to be uniform? Questions like these may provide insightful instructions on how to collect labels for class-imbalanced learning, which I find useful for many real-world applications. --------- Updates: The authors response has addressed my concerns. I recommend acceptance for this paper.

Correctness: The theoretical analysis is correct and the empirical evaluations are solid in this paper.

Clarity: The paper is written excellent.

Relation to Prior Work: The paper cited adequate related works and clearly describe its own novelties.

Reproducibility: Yes

Additional Feedback:


Review 4

Summary and Contributions: The paper studies the problem of long-tailed recognition. It theoretically and empirically demonstrates that (1) in most of the cases, semi-supervised learning can help with long-tailed recognition, unless the unlabeled data is heavily imbalanced (2) the representation learned by self-supervised pretraining can improve class-imbalanced learning. The proposed self-supervised imbalanced learning framework (SSP) achieves new SOTA on a number of long-tailed recognition benchmarks: CIFAR-10-LT, CIFAR-100-LT, ImageNet-LT and iNaturalist 2018.

Strengths: (1) If I understand correctly, the proposed SSP method follows a similar paradigm with cRT[24]: learning the representation and classifier in different stages. While cRT[24] relies on imbalanced labels to learn the representation, this paper proposes to pretrain the feature representation in a self-supervised manner. It shows that the representation learned by self-supervised pretraining can improve class-imbalanced learning (or at least is complementary to other SOTA methods, as shown in table 3), which is very interesting. (2) The paper shows that, in most of the cases, leveraging unlabeled data in a semi-supervised setting can improve long-tailed recognition, unless the unlabeled data is heavily imbalanced (p_{U} > 50). Although the experiments are conducted on small-scale benchmarks only (e.g. CIFAR-10-LT, SVHN-LT), the finding can still provide some guidance for collecting unlabeled data for semi-supervised learning, especially when the available data is imbalanced.

Weaknesses: The paper claims that, the semi-supervised experiment demonstrates the value of labeled data. This claim looks odd. The labeled data is the “control variable” (remains unchanged) of the experiment, while the unlabeled data is the ‘independent variable’ of the experiment. I believe any conclusion drawn from the experiment should be associated with the “independent variable”. In other words, the experiment only demonstrates the value of unlabeled data for long-tailed recognition.

Correctness: The proposed method is technically sound and interesting. The claim that the semi-supervised experiment demonstrates the value of labeled data looks odd. Please refer to the weakness part.

Clarity: The paper is well written.

Relation to Prior Work: I feel like the proposed SSP method shares some similarities with cRT[24]. It would be better if more discussion on the difference between cRT[24] and the proposed SSP is included.

Reproducibility: Yes

Additional Feedback: -------------------------- Post-rebuttal: The rebuttal resolves some of my concerns. I’d like to keep my score (6), and vote for acceptance.

[Author Response · NeurIPS 2020]

We thank all the reviewers for acknowledging the contributions of our work and providing insightful comments and
suggestions. In the following, we address all the concerns of each reviewer in detail.

**[Reviewer #1]**

**Reflect on the pseudo-label algorithm.** We agree that a poor self-training procedure might reinforce errors. However,
as we demonstrated both theoretically and empirically, given a fairly good base classifier, more relevant unlabeled
data is helpful (cf. discussions in Section 2.1, L93, and results in Table 1), regardless of the data imbalanceness as
well as base SSL methods. We remark that here, our main aim is to explore how semi-supervised techniques help in
imbalanced learning. Further ablation studies on different SSL algorithms are also provided in appendices to give a
complete picture. Altogether, we believe that these justify the value of SSL and our claims. We plan to include more
discussions in the main paper to reflect on the usage of different SSL algorithms.

**Compare to one other algorithm and ignore others.** We would like to clarify that we actually did a comprehensive
study (main paper and appendices) on the effect of the proposed techniques through **5 dimensions**: different (1) datasets,
(2) imbalance ratio, (3) unlabeled data imbalance ratio, (4) balanced (imbalanced) training strategy, and (5) SSL method.
The key point is to demonstrate that SSL as a technique is generally beneficial for both representative non-imbalanced
methods (CE in Table 1) and imbalanced methods (LDAM-DRW). We believe the extensive tests over the above
dimensions, together with consistent improvements in performance, confirm the impact of our contributions.

**More detailed related work.** We agree with you that the section on related work is relatively short. While we include
most of the relevant literature, the discussions are not elaborated enough. This is largely due to the space limit. We will
definitely include more details to provide an informed picture for the audience in the revised version.

**[Reviewer #2]**

**Definition of relevance score.** Thank you for pointing this out. We did not define it clearly in the main text, but have
elaborated it in Appendix D.2 (from L541). We will bring the detailed setups to the main paper in the revised version.

**Theorems w.r.t. relevance score.** This is a good point. In our paper, to motivate our study and provide insights, the
theorems mainly consider the imbalanceness for labeled & unlabeled data. When data relevance comes into the picture,
it might not be easy to obtain a clean and rigorous mathematical formalization. We believe the current theorems are of
their own interests, and should be insightful enough in understanding key components in imbalanced learning. On the
other hand, including other components such as relevance, is certainly an interesting future direction to consider.

**Self-supervision vs. irrelevant unlabeled data.** Section 3 mainly motivates SSP by presenting the potential practical
issues of unlabeled data. For self-supervised methods, we would like to clarify that our key theme is to investigate the
two perspectives on the dilemma of imbalanced labels. Hence, we decompose the usage of semi- & self-supervised
learning, and analyze each of them *separately* on how they improve imbalanced learning. As such, in our experiments
of SSP, we do not consider extra unlabeled data. However, we do appreciate the idea of combining both to further
improve results. As you noted, one should be always aware of the potential risk of irrelevant unlabeled data in practice.

**Title & Theorems towards complex models.** We intended to let the title reflect the main perspectives of our investiga-
tion on the imbalanced labels: the positive and the negative side of the dilemma. We certainly welcome any suggestions.
For the theorems, these are excellent points. We would be interested in a fully general theorem. A thorough analysis of
every aspect is likely hard and beyond the scope of current manuscript, but is definitely an important future direction.

**[Reviewer #3]**

**More analysis on different label distributions.** This is a great suggestion, and we believe a comprehensive study is
broadly valuable in real-world applications. In fact, as a first step, we have made efforts to address it by considering
different imbalance type of the labeled data. The main paper focuses on long-tailed distribution which is most common
in literature, but we also further study the step imbalance distribution in Appendix F.3. We remark that within both
imbalance types, we investigate various distributions by changing the imbalance ratio. The conclusions are consistent
across those studies. We will include a more clear summary of our results in the revised version, and remark on the
practical importance of further analysis on different imbalanced distributions.

**[Reviewer #4]**

**Claim on SSL demonstrating the value of labeled data.** This is an interesting perspective. To clarify, let us elaborate
our viewpoint provided in the framework. We regard the labeled data as providing useful information and positive
guidance for the unlabeled data, i.e., offering pseudo-label. This in return, as we consistently demonstrated, helps the
overall learning. In this sense, the labels indeed contain positive value. Further, we remark that the labeled data may not
be viewed solely or exactly as "unchanged controlled variable". In our experiments, we also actively study the amount
of labeled & unlabeled data (Appendix E.3), i.e., the amount of "value" or supervision the label provides. We will
provide a clear summary to distinguish our perspective on the positive value of the imbalanced label.

**Discussion between cRT and the proposed SSP.** Your general feeling is correct: in a sense, SSP also aims to learn
better representations that are more agnostic to the imbalanced label bias. As mentioned in our response to Reviewer
#1, with more space in the next revision, we will definitely include a more detailed discussion in the related work.

[Meta-Review · NeurIPS 2020]

This work considers the class-imbalance setting, specifically the potential benefits of using semi-supervised/self-supervised learning. Based on theoretical observations regarding unlabeled data in this setting, a pseudo-labeling strategy is proposed for training and pre-training, analyzed, and thoroughly evaluated. Even after rebuttal and discussion, there remained some remaining suggestions around additional citations, etc. (as this is a well-established area), but these were not crucial in my opinion. However, the analysis and empirical findings were considered important by all the reviewers (especially when including the appendices) and there was unanimous support for accepting.